# D²-LoRA: A Synergistic Approach to Differential and Directional Low-Rank Adaptation

## Abstract

We systematically investigate the parameter-efficient fine-tuning design space under practical data and compute constraints, yielding D²-LoRA—a method that achieves **76.4%** average accuracy across eight QA/RC benchmarks using **only 5k training samples per task** and **two epochs**, while preserving algebraic mergeability at inference with near-exact numerical equivalence. The core idea combines a signed low-rank residual $\Delta W = \frac{\alpha}{r}(A_+ B_+ - \tau A_- B_-)$ that admits both additive and subtractive updates with a train-time columnwise projection that constrains each column to its original norm. After training, the adapter collapses into a single matrix $\widehat{W}$ incurring zero additional inference latency. Compared with LoRA, D²-LoRA achieves a **+2.2 pp** improvement on average; at matched parameter counts (LoRA at rank $2r$ vs. D²-LoRA at rank $r$), the improvement is **+1.6 pp**, confirming that gains stem from architectural innovations rather than increased parameterization. Compared with DoRA, it matches or exceeds performance on most tasks. Beyond QA/RC, D²-LoRA improves generative tasks (**+1.2 pp** ROUGE-L, **+1.1%** win rate) and exhibits **36% lower training volatility**. The merge preserves numerical fidelity (mean gap $\approx 0.03$ pp) and recovers $\sim$**1.91×** evaluation throughput. Training overhead is **19%**—comparable to DoRA—and decreases with longer input sequences. We provide a geometric analysis explaining how the proposed projection stabilizes training, together with ablation studies isolating the contribution of each design component.

## 1 Introduction

Fine-tuning large-scale transformers is often constrained by practical resource limitations rather than methodological considerations. Parameter-efficient fine-tuning (PEFT) addresses the common scenario where practitioners have access to limited compute—a single GPU over a weekend rather than a cluster for a month. LoRA (Hu et al., 2022) exemplifies practical efficiency by learning a low-rank residual that can be merged into the base weights, thereby preserving inference efficiency. DoRA (Liu et al., 2024) adopts an alternative strategy: updating only directional components while keeping magnitudes fixed, which improves training stability but may constrain representational capacity at low ranks.

We investigate the low-data, low-rank regime and identify two key desiderata for effective adaptation: (i) regulation of column-wise magnitudes during optimization, and (ii) preservation of both additive and subtractive representational capacity. D²-LoRA satisfies both requirements. It introduces a signed residual—a second branch scaled by $\tau$ that acts in opposition to the first—while a train-time projection constrains $W_0 + \Delta W$ to lie on the product of spheres defined by the original column norms. Since the projection operates only during training, D²-LoRA retains the key advantage of LoRA: a single merged weight matrix at inference with no additional computational overhead.

**Contributions.** **(1) Mergeable design space.** We propose a PEFT formulation that *strictly subsumes* LoRA and can recover a DoRA-like variant by toggling two orthogonal dimensions: signed residuals and directional projection. **(2) Theory that predicts stability.** A geometric analysis reveals how the proposed projection removes radial gradient components and enforces Lipschitz continuity—precisely the stability needed in small-data, low-rank regimes. Empirically, D²-LoRA exhibits **36% lower loss volatility** than LoRA. **(3) Comprehensive evaluation.** Under a strict

Table 1: PEFT design choices and their implications.

| Method | Mergeable | Norm control | Small-data stability | Rank expressivity |
|---|---|---|---|---|
| LoRA | ✓ | | △ | $r$ |
| DoRA | ✓ | ✓ | △ | $r$ |
| **D$^2$-LoRA** | ✓ | ✓ | ✓ | $2r$ |

training budget (5k samples, two epochs) across eight QA/RC benchmarks, two generative tasks, and two backbones, D$^2$-LoRA achieves **76.4%** macro accuracy (**+2.2 pp** over LoRA) with consistent generative gains (**+1.2 pp** ROUGE-L, **+1.1%** win rate), while maintaining near-exact merge fidelity and recovering $\sim 1.91\times$ inference throughput. At matched parameters, the **+1.6 pp** advantage confirms that architectural design outweighs capacity scaling. **(4) Attribution-aware ablations.** We dissect the performance gains by attributing improvements to each factor: the signed negative branch, module targeting, rank, scoring metric, and the fixed $\tau$. We also provide direct evidence of stability improvements via quantitative plots.

## 2 RELATED WORK

**Adapters and prompts.** Adapters introduce small trainable bottlenecks (Houlsby et al., 2019; Chen et al., 2022; Sung et al., 2022; Lu et al., 2024); BitFit updates biases (Ben Zaken et al., 2022). Prompt-based approaches tune continuous prompts (Lester et al., 2021; Li & Liang, 2021; Jia et al., 2022).

**Low-rank updates.** LoRA (Hu et al., 2022) trains mergeable low-rank residuals; QLoRA (Dettmers et al., 2023) trades precision for memory; AdaLoRA (Zhang et al., 2023) adapts rank budgets; NOLA compresses adapters (Koohpayegani et al., 2024). These methods do not directly constrain per-column magnitudes during training. We focus comparisons on LoRA and DoRA as they represent the two dominant paradigms (unconstrained vs. directionally-constrained updates) and share D$^2$-LoRA's key property of algebraic mergeability. Methods like QLoRA address orthogonal concerns (memory efficiency) and AdaLoRA requires additional hyperparameter tuning for rank allocation, making direct comparison less informative for isolating the benefits of signed residuals and directional projection.

**Directional factorization.** DoRA (Liu et al., 2024) decouples magnitude and direction, updating the latter. D$^2$-LoRA extends this by restoring residual expressivity through signed branches while maintaining directional constraints. While Differential Transformer (Ye et al., 2025) applies differential principles in attention space, D$^2$-LoRA operates in parameter space for adaptation without architectural changes.

## 3 D$^2$-LoRA MODEL ARCHITECTURE

**Preliminaries.** Let $W_0 \in \mathbb{R}^{d_\text{out} \times d_\text{in}}$ be a frozen linear map with bias $b$. For $X \in \mathbb{R}^{d_\text{out} \times d_\text{in}}$, define $\|X\|_{2,\text{col}} \in \mathbb{R}^{d_\text{in}}$ by $\left(\|X\|_{2,\text{col}}\right)_j = \|X^{[:,j]}\|_2$. We follow the `nn.Linear` convention that stores weights transposed relative to the forward; formulas below are presented in $(d_\text{out}, d_\text{in})$ for clarity.

**Overview.** Figure 1 summarizes the architecture at a glance: a signed low-rank residual $\Delta W = \frac{\alpha}{r}(A_+ B_+ - \tau A_- B_-)$ is composed with a *train-time* columnwise directional projection that preserves the baseline magnitudes of $W_0$; at inference we merge into a single matrix to avoid any latency.

**Signed low-rank residual.** We maintain two factor pairs $A_\pm \in \mathbb{R}^{d_\text{in} \times r}$, $B_\pm \in \mathbb{R}^{r \times d_\text{out}}$ and define

$$\Delta W^\top = \frac{\alpha}{r}\left(A_+ B_+ - \tau A_- B_-\right), \tag{1}$$

with a scalar $\tau$ (fixed by default), yielding $\text{rank}(\Delta W) \leq 2r$.

**Directional projection (train time only).** Let $\mathbf{m} = \|W_0\|_{2,\text{col}}$, $\mathbf{d} = \|W_0 + \Delta W\|_{2,\text{col}}$ with entries $d_j$, and define $\mathbf{d}_\varepsilon = \max(\mathbf{d}, \varepsilon \mathbf{1})$ (i.e., $d_j^\varepsilon = \max(d_j, \varepsilon)$). We normalize columns to the baseline

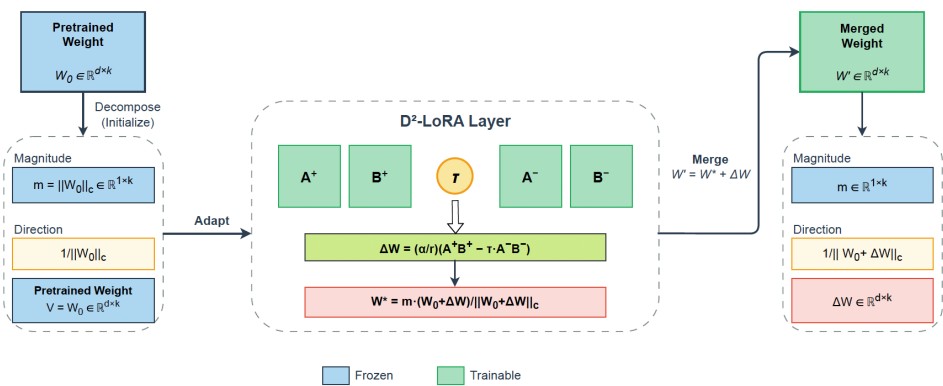

Figure 1: **Overview of D$^2$-LoRA.** Left: columnwise magnitude $\mathbf{m} = \|W_0\|_{2,\text{col}}$ from the pretrained weight $W_0$. Middle: differential signed low-rank residual with scale $\tau$ and a *train-time* directional projection that preserves the column norms of $W_0$. Right: inference uses the merged weight $\widehat{W} = W^\star + \Delta W$, so latency equals a single linear layer.

---

**Algorithm 1** D$^2$-LoRA for one linear projection

---

**Require:** Frozen $W_0$, rank $r$, scale $\alpha$, fixed $\tau, \varepsilon > 0$
1: Initialize $A_+, A_-, B_+, B_-$ (smaller std for $A_-$); set $\mathbf{m} = \|W_0\|_{2,\text{col}}$
2: **for** each step **do**
3:     $\Delta W^\top \leftarrow \frac{\alpha}{r}(A_+ B_+ - \tau A_- B_-)$
4:     $\mathbf{d} \leftarrow \|W_0 + \Delta W\|_{2,\text{col}}; \quad \mathbf{d}_\varepsilon \leftarrow \max(\mathbf{d}, \varepsilon \mathbf{1}); \quad W^\star \leftarrow (W_0 + \Delta W) \operatorname{diag}(\mathbf{m} /\!/ \mathbf{d}_\varepsilon)$
5:     $y \leftarrow x W^{\star\top} + \mathcal{D}(x) \Delta W^\top + b$          (residual-path dropout only)
6: **end for**
7: **Merge:** $\widehat{W} \leftarrow W^\star + \Delta W$

---

magnitudes:
$$W^\star = (W_0 + \Delta W) \cdot \operatorname{diag}(\mathbf{m} /\!/ \mathbf{d}_\varepsilon). \tag{2}$$

Only the residual path receives input dropout; the backbone path remains untouched. At inference we *merge* $W^\star$ and $\Delta W$ as

$$\widehat{W} = W^\star + \Delta W, \tag{3}$$

thereby the latency and FLOPs are equal to those of a single linear layer.

**Why add $\Delta W$ twice?** During training, $y = x W^{\star\top} + \mathcal{D}(x) \Delta W^\top + b$ creates two signal paths: the *directional branch* ($W^\star$) provides norm-controlled updates, while the *residual branch* ($\Delta W$) provides unconstrained capacity with dropout. At inference ($\mathcal{D} = \text{Id}$), both contribute equally via $\widehat{W} = W^\star + \Delta W$. Ablations confirm this dual-path design is essential: removing the residual branch reduces accuracy by $-0.6$ pp (Table 6).

**Configuration Knobs.** Disabling equation 2 recovers the original LoRA formulation; setting the minus branch to zero yields a DoRA-like variant. The primary design axes are therefore: the signed residual (enabled/disabled), the directional projection (enabled/disabled), and the scalar parameter $\tau$.

**Rationale for small-minus initialization.** Our initialization strategy directly impacts the optimization trajectory. With $\Delta W^\top = \frac{\alpha}{r}(A_+ B_+ - \tau A_- B_-)$ and the directional projection $P$ in equation 2, we initialize $A_+ \sim \mathcal{N}(0, \sigma^2 I)$ and $A_- \sim \mathcal{N}(0, 0.01\sigma^2 I)$, while $B_\pm$ start at zero. This $10\times$ variance reduction for the negative branch serves three critical purposes: (1) it prevents destructive cancellation $W_0 + \Delta W \approx 0$ that would render the Jacobian of $P$ ill-conditioned through the $1/\|W_0 + \Delta W\|$ term, (2) it establishes a natural learning curriculum where the model first learns additive patterns before discovering which features to suppress, and (3) it acts as an implicit gating mechanism that gradually activates the negative branch as training progresses.

**Empirical validation.** Table 16 (Appendix I.1) demonstrates that standard initialization ($A_- = A_+$) causes training instability on 3/8 datasets (NaN loss within 50 steps), while our $10\times$ reduction achieves stable training on all datasets with the highest accuracy (76.4%). This confirms that the asymmetric initialization is necessary for stable optimization, not merely a design choice.

## 4  THEORY

### 4.1  D²-LoRA STABILITY

**Proposition 1** (Expressivity gap of signed low-rank updates). *For rank $r$, LoRA admits updates $\Delta W$ with $\operatorname{rank}(\Delta W) \leq r$. By contrast, D²-LoRA produces $\Delta W^\top = \frac{\alpha}{r}(A_+ B_+ - \tau A_- B_-)$ with $\operatorname{rank}(\Delta W) \leq 2r$. Moreover, for any rank-$r$ matrix $M$, there exist $(A_\pm, B_\pm, \tau)$ such that $\Delta W = M$ (e.g., $A_- = 0$), and for generic $(A_\pm, B_\pm)$ with $\tau \neq 0$ the column space of $\Delta W$ equals the direct sum of the column spaces of $A_+ B_+$ and $A_- B_-$ up to dimension $2r$.*

*Proof sketch.* Write $\Delta W^\top = \frac{\alpha}{r}(A_+ B_+ - \tau A_- B_-)$ with $\operatorname{rank}(A_\pm B_\pm) \leq r$. Then $\operatorname{rank}(\Delta W) \leq \operatorname{rank}(A_+ B_+) + \operatorname{rank}(A_- B_-) \leq 2r$. For any rank-$r$ matrix $M$, set $A_- = 0$ and choose $A_+ B_+ = \frac{r}{\alpha} M^\top$ to recover $\Delta W = M$. For generic $(A_\pm, B_\pm)$ with $\tau \neq 0$, the column spaces of $A_+ B_+$ and $A_- B_-$ are in general position, giving typical rank $2r$. A complete proof is provided in Appendix E. $\square$

**Geometric role of $\tau$.** Let $U_\pm = \operatorname{col}(A_\pm B_\pm)$ and let $\theta_1(U_+, U_-)$ denote the smallest principal angle. Scaling by $\tau$ controls the balance between constructive and destructive interference of $U_+$ and $U_-$. A balanced choice $\tau \approx 1$ prevents domination by either branch, while $\tau \ll 1$ collapses toward LoRA (positive branch only) and $\tau \gg 1$ magnifies cancellation and can ill-condition early training. This explains the broad optimum $\tau \in [0.5, 1.0]$ observed in Table 7.

The projection equation 2 constrains each input column to a sphere of radius $m_j = \|W_0^{[:,j]}\|_2$.

**Proposition 2** (Norm preservation). *For any $\varepsilon > 0$, $\|W^{\star[:,j]}\|_2 = m_j$ whenever $\|W_0^{[:,j]} + \Delta W^{[:,j]}\|_2 \geq \varepsilon$, and $\|W^{\star[:,j]}\|_2 \leq m_j$ otherwise.*

*Proof sketch.* For column $j$, let $u_j = (W_0 + \Delta W)^{[:,j]}$ and $d_j^\varepsilon = \max(\|u_j\|_2, \varepsilon)$. Since $W^{\star[:,j]} = \frac{m_j}{d_j^\varepsilon} u_j$, we have $\|W^{\star[:,j]}\|_2 = \frac{m_j}{d_j^\varepsilon}\|u_j\|_2$, which equals $m_j$ if $\|u_j\|_2 \geq \varepsilon$ and is $\leq m_j$ otherwise. See Appendix E for the complete proof. $\square$

**Lemma 1** (Smoothness and Lipschitz control). *The map $P : \Delta W \mapsto W^\star$ is $C^1$ and locally Lipschitz with constant $C(\mathbf{m}, \varepsilon)$, so $\|P(\Delta W_1) - P(\Delta W_2)\|_F \leq C \|\Delta W_1 - \Delta W_2\|_F$.*

*Proof sketch.* Columnwise $\phi(u) = u/\|u\|_\varepsilon$ is $C^1$ on $\{\|u\| > \varepsilon\}$ with Jacobian $D\phi(u) = \|u\|^{-1}\left(I - \frac{uu^\top}{\|u\|^2}\right)$ and operator norm $\leq 1/\varepsilon$, and linear with Lipschitz constant $1/\varepsilon$ when $\|u\| \leq \varepsilon$. Composing the affine map $\Delta W \mapsto W_0 + \Delta W$ with $\phi$ and scaling by $\operatorname{diag}(\mathbf{m})$ yields local Lipschitz continuity of $P$ with constant $C(\mathbf{m}, \varepsilon)$. Full details appear in Appendix E. $\square$

**Benefits of directional constraints in low-data regimes.** Let $\mathcal{M} = \prod_{j=1}^{d_{\text{in}}} S^{d_{\text{out}}-1}(m_j)$ denote the product of spheres defined by $\mathbf{m}$. The projection $P$ restricts the training dynamics to $\mathcal{M}$ while the residual branch provides additive and subtractive degrees of freedom via equation 1. When $n \ll d_{\text{out}} d_{\text{in}}$, overparameterized radial modes amplify variance; $P$ eliminates these modes by zeroing radial gradient components and aligning updates to the tangent bundle $T\mathcal{M}$. The resulting reduction in effective degrees of freedom provides implicit regularization. Empirically, this manifests as smoother loss trajectories and improved training stability at low rank, with the most pronounced benefits on HellaSwag and WinoGrande.

**Merge-time equivalence.** With dropout disabled, $x(W^\star + \Delta W)^\top + b = xW^{\star\top} + x\Delta W^\top + b$; merged and unmerged evaluation are thus identical up to numerical roundoff, matching our measurements.

## 4.2 ON CHOOSING THE SCALE $\tau$

Under independent, zero-mean isotropic initializations for $(A_\pm, B_\pm)$, the two branches are uncorrelated so that

$$\mathbb{E}\|\Delta W\|_F^2 = \frac{\alpha^2}{r^2}\Big(\mathbb{E}\|A_+ B_+\|_F^2 + \tau^2\,\mathbb{E}\|A_- B_-\|_F^2\Big).$$

Thus $\tau$ controls the energy injected by the negative branch. In the small-step regime with directional projection $P$ (equation 2), stability depends on the radial component of the update; balancing the branch energies ($\tau \approx 1$) reduces anisotropy before projection, while $\tau \ll 1$ under-utilizes subtractive directions and $\tau \gg 1$ increases cancellation risk. This yields a model-agnostic prior $\tau \in [0.5, 1.0]$, consistent with Table 7.

**Practical guidance for $\tau$ selection.** Table 7 shows that performance varies by at most 1.1 pp across the tested range ($\tau \in \{0.25, 0.5, 1.0\}$), indicating low sensitivity. We recommend:

- **Default:** $\tau = 0.5$ provides robust performance across models and tasks.
- **If tuning budget allows:** Try $\tau \in \{0.5, 1.0\}$ (2 runs) and select based on validation loss.
- **Model-specific trends:** Larger models (Qwen2.5-7B) slightly prefer $\tau = 0.5$; smaller models (Llama-3.2-3B) slightly prefer $\tau = 1.0$.

Unlike rank or target module selection, $\tau$ tuning provides marginal gains and can be safely fixed to 0.5 without significant performance loss.

## 5 EXPERIMENTAL SETUP

**Backbones.** *Llama-3.2-3B-Instruct* and *Qwen2.5-7B-Instruct*; all backbone weights frozen.

**Tasks.** Eight QA/RC datasets: BoolQ (Clark et al., 2019), CommonsenseQA (Talmor & Berant, 2019), RACE (Lai et al., 2017), HellaSwag (Zellers et al., 2019), WinoGrande (M) (Sakaguchi et al., 2020), ARC-Easy/Challenge (Clark et al., 2018), and OpenBookQA (Mihaylov et al., 2018). We use a single multiple-choice template and consistent tokenization across methods.

**Budget and preprocessing.** Training uses at most 5,000 samples per dataset; validation uses 500 instances (ARC-Challenge: 299). The primary metric is option accuracy under conditional log-likelihood (CLL); we also report a generation-based scorer for comparison with identical decoding/truncation settings.

**Defaults.** Unless specified: rank $r = 32$ with $\alpha = 64$ for Llama-3.2-3B-Instruct, and rank $r = 16$ with $\alpha = 32$ for Qwen2.5-7B-Instruct (so that $\alpha/r = 2$ on both backbones); a single global $\tau = 0.5$ fixed a priori for the main tables (ablations for $\tau \in \{0.25, 0.5, 1.0\}$ are in Sec. 5.4); residual-path input dropout 0.1, matrix-dropout off, AdamW with cosine LR, batch 16 with grad-accum 2, two epochs. We adapt $q, k, v, o$ unless noted otherwise.

### 5.1 MAIN RESULTS

Table 2 summarizes the main results across both backbones. On Llama-3.2-3B-Instruct, D$^2$-LoRA achieves consistent improvements across all eight datasets, attaining a macro average of 76.4%—a **+2.2 pp** improvement over LoRA (74.2%). The largest gains occur on tasks requiring fine-grained linguistic discrimination: WinoGrande improves by **+4.8 pp** (64.0% vs. 59.2%), and HellaSwag by **+4.8 pp** (82.4% vs. 77.6%). These results suggest that the combination of signed residuals and directional constraints captures semantic patterns that standard LoRA fails to model effectively.

On Qwen2.5-7B-Instruct, D$^2$-LoRA achieves the highest macro average (85.7%) with competitive performance across all tasks, demonstrating that the method generalizes across model scales.

**Head-to-head comparison (Llama-3.2-3B-Instruct).** Against LoRA: **8 wins, 0 ties, 0 losses**. Against DoRA: **5 wins, 1 tie, 2 losses**. Consistency across diverse tasks supports the design choices.

*Protocol note:* We employ a single global $\tau = 0.5$ across all experiments to ensure fair comparison. While $\tau = 1.0$ yields marginal improvements (+0.74 pp, Sec. 5.4), we prioritize generalization over task-specific tuning.

Table 2: Main results: Accuracy (%) on 8 QA/RC tasks with CLL scoring. Best per backbone in **bold**.

| Dataset | Llama-3.2-3B-Instruct (2 epochs) | | | Qwen2.5-7B-Instruct (1 epoch) | | |
|---|---|---|---|---|---|---|
| | LoRA | DoRA | $D^2$-LoRA | LoRA | DoRA | $D^2$-LoRA |
| ARC-Challenge | 72.6 | 72.2 | **73.9** | **89.6** | 88.3 | **89.6** |
| ARC-Easy | 87.2 | **87.6** | 87.4 | **96.2** | **96.2** | **96.2** |
| BoolQ | 62.6 | 61.8 | **65.6** | 82.0 | 84.2 | **85.2** |
| CommonsenseQA | 73.4 | 74.8 | **75.4** | **82.8** | 81.8 | 82.2 |
| HellaSwag | 77.6 | 81.8 | **82.4** | 90.2 | 90.0 | **91.0** |
| OpenBookQA | 81.8 | **83.4** | 82.6 | 87.0 | 88.0 | **89.0** |
| RACE | 79.0 | **79.6** | **79.6** | **90.2** | **90.2** | 89.6 |
| WinoGrande | 59.2 | 62.6 | **64.0** | **63.6** | 61.6 | 63.0 |
| **Average** | 74.2 | 75.5 | **76.4** | 85.2 | 85.0 | **85.7** |

Table 3: CLL vs generation (Llama-3.2-3B-Instruct, $r{=}32$). Entries show CLL$-$Gen in pp.

| Dataset | LoRA | DoRA | $D^2$-LoRA |
|---|---|---|---|
| ARC-Challenge | +2.0 | +0.3 | +0.7 |
| ARC-Easy | +0.6 | +0.4 | +0.2 |
| BoolQ | +0.6 | $-1.4$ | +1.2 |
| CommonsenseQA | +0.0 | +0.0 | +1.2 |
| HellaSwag | +2.6 | +5.4 | +5.6 |
| OpenBookQA | $-0.4$ | +0.0 | +0.6 |
| RACE | $-0.2$ | +0.6 | +0.4 |
| WinoGrande | +1.4 | +2.0 | +0.4 |
| **Average** | **+0.9** | **+0.9** | **+1.3** |

## 5.2 CLL VS. GENERATION SCORING

CLL consistently outperforms generation-based scoring by $+0.9$–$+1.3$ pp across methods (Table 3), with the largest gap on HellaSwag (+5.6 pp). $D^2$-LoRA shows the largest CLL advantage (+1.3 pp), suggesting its representations are well-suited for discriminative evaluation.

## 5.3 MERGE EQUIVALENCE AND THROUGHPUT

Practical deployment requires merging the adapter into base weights without degrading performance. $D^2$-LoRA maintains near-exact numerical equivalence between merged and unmerged configurations. On Llama-3.2-3B-Instruct, the mean accuracy difference is 0.03 pp, with a maximum deviation of 0.7 pp on ARC-Challenge—well within typical evaluation variance (Table 4). The train-time projection does not perturb the algebraic merge: with dropout disabled, $x(W^\star + \Delta W)^\top = xW^{\star\top} + x\Delta W^\top$ holds up to floating-point precision.

## 5.4 ABLATIONS

**Rank.** Reducing $r$ from 32 to 16 costs $-2.2$ pp on average; $r{=}8$ retains 73.6% (Table 5). A practical threshold emerges around $r{=}16$ under our budget: most of the signed-branch expressivity is realized by $r{=}16$, with diminishing returns beyond. **Target modules.** Adapting $q, v, o$ (dropping $k$) saves $\sim 25\%$ parameters for $-0.3$ pp; restricting to $q, k$ halves parameters but hurts accuracy (Table 5). **Negative branch.** Co-training the $(-)$ branch matters; disabling or gradient-detaching underperforms (Table 6). **Matrix dropout.** Helps BoolQ but hurts ARC-Challenge on average; we keep it off by default (Table 6). **Fixed $\tau$.** Macro averages for $\tau \in \{0.25, 0.5, 1.0\}$ (Llama-3.2-3B-Instruct) and $\{0.5, 1.0\}$ (Qwen2.5-7B-Instruct) are reported jointly in Table 7; the figure visualizes the same numbers.

Table 4: $D^2$-LoRA merge equivalence across backbones. Accuracy (%) and post-merge evaluation speedup. Speedup is computed as (unmerged $D^2$-LoRA evaluation time) / (merged $D^2$-LoRA evaluation time), i.e., comparing merged vs. unmerged configurations of the *same* $D^2$-LoRA model. This measures the inference benefit of merging, not comparison against other methods.

| Backbone | Dataset | Unmerged | Merged | Speedup |
|---|---|---|---|---|
| Llama-3.2-3B-Instruct | ARC-Challenge | 73.9 | 73.2 | 1.89× |
| | ARC-Easy | 87.4 | 87.4 | 1.87× |
| | BoolQ | 65.6 | 65.6 | 1.98× |
| | CommonsenseQA | 75.4 | 74.8 | 1.82× |
| | HellaSwag | 82.4 | 82.6 | 2.03× |
| | OpenBookQA | 82.6 | 83.0 | 1.85× |
| | RACE | 79.6 | 79.8 | 2.02× |
| | WinoGrande | 64.0 | 64.2 | 1.85× |
| | **Average** | **76.36** | **76.33** | 1.91× |
| Qwen2.5-7B-Instruct | ARC-Challenge | 89.6 | 89.6 | 2.31× |
| | ARC-Easy | 96.2 | 96.2 | 2.30× |
| | BoolQ | 85.2 | 84.6 | 2.01× |
| | CommonsenseQA | 82.2 | 81.8 | 2.31× |
| | HellaSwag | 91.0 | 90.8 | 1.90× |
| | OpenBookQA | 89.0 | 88.8 | 2.29× |
| | RACE | 89.6 | 89.2 | 1.71× |
| | WinoGrande | 63.0 | 61.8 | 2.27× |
| | **Average** | **85.4** | **85.3** | 2.14× |

Table 5: Ablation studies (Llama-3.2-3B-Instruct, CLL scoring): (a) effect of rank $r$; (b) effect of target modules ($r$=32).

(a) Effect of rank $r$

| Dataset | $r$=32 | $r$=16 | $r$=8 |
|---|---|---|---|
| ARC-Challenge | **73.9** | 72.2 | 73.2 |
| ARC-Easy | **87.4** | **87.4** | 86.6 |
| BoolQ | **65.6** | 59.2 | 60.8 |
| CommonsenseQA | **75.4** | 75.0 | 74.0 |
| HellaSwag | **82.4** | 78.6 | 74.8 |
| OpenBookQA | 82.6 | **82.8** | 82.6 |
| RACE | **79.6** | 79.2 | 79.0 |
| WinoGrande | **64.0** | 58.8 | 57.8 |
| **Average** | **76.4** | 74.2 | 73.6 |

(b) Effect of target modules

| Dataset | $q, k, v, o$ | $q, v, o$ | $q, k$ |
|---|---|---|---|
| ARC-Challenge | 73.9 | 72.6 | **74.2** |
| ARC-Easy | 87.4 | **87.6** | 86.0 |
| BoolQ | **65.6** | 61.6 | 53.0 |
| CommonsenseQA | **75.4** | 74.4 | 68.6 |
| HellaSwag | **82.4** | 81.8 | 67.4 |
| OpenBookQA | **82.6** | 82.2 | 79.2 |
| RACE | **79.6** | 79.2 | 78.2 |
| WinoGrande | 64.0 | **64.4** | 56.4 |
| **Average** | **76.4** | 75.5 | 70.4 |

**Asymmetric rank allocation.** We also explore asymmetric ranks for the $\pm$ branches (Appendix L, Table 19). At fixed total parameter budget (1.58M), mild asymmetry ($r_+ = 24$, $r_- = 8$) yields marginal improvement (+0.2 pp), but strong asymmetry ($r_+ = 28$, $r_- = 4$ or $r_+ = 8$, $r_- = 24$) degrades performance. Symmetric ranks ($r_+ = r_- = 16$) achieve near-optimal results and are simpler to configure.

**MLP layers.** We investigate adapting MLP layers (gate, up, down projections) in addition to attention. Appendix J (Table 17) shows that adding MLP layers yields diminishing returns: +0.4 pp for gate+up (2.37M params), +0.7 pp for all MLP layers (3.16M params). Given the 3× parameter increase for only +0.8 pp improvement, attention-only adaptation provides the best efficiency-performance tradeoff.

## 5.5 PARAMETER-MATCHED COMPARISON

A critical question is whether $D^2$-LoRA's gains stem from architectural innovations or simply from increased parameterization. We compare $D^2$-LoRA at $r$=32 against LoRA and DoRA at $r$=64, ensuring identical parameter budgets ($\sim$1.58M).

Table 6: Ablation studies (Llama-3.2-3B-Instruct, $r$=32, CLL scoring): (a) effect of the negative branch; (b) matrix dropout. **Note on identical results:** "Minus off" and "Detach minus grad" yield identical results because both configurations prevent the negative branch from contributing to the update. With "Minus off," $A_- B_- = 0$ by construction. With "Detach minus grad," gradients are zeroed, but since $B_-$ is initialized to zero and receives no gradient updates, $A_- B_- = 0$ throughout training. This is *not* a typo—it confirms that the negative branch's contribution requires active gradient flow.

| (a) Negative branch | | | | (b) Matrix dropout | | |
|---|---|---|---|---|---|---|
| Dataset | Full | Minus off | Detach minus grad | Dataset | $p$=0 | $p$=0.10 |
| ARC-Challenge | **73.9** | 73.6 | 73.6 | ARC-Challenge | **73.9** | 72.2 |
| ARC-Easy | **87.4** | 87.2 | 87.2 | ARC-Easy | 87.4 | **87.8** |
| BoolQ | 65.6 | **66.0** | 66.0 | BoolQ | 65.6 | **68.0** |
| CommonsenseQA | **75.4** | 74.6 | 74.6 | CommonsenseQA | **75.4** | 74.6 |
| HellaSwag | **82.4** | 81.6 | 81.6 | HellaSwag | **82.4** | 81.6 |
| OpenBookQA | **82.6** | 82.4 | 82.4 | OpenBookQA | **82.6** | 82.0 |
| RACE | **79.6** | 79.0 | 79.0 | RACE | **79.6** | 79.0 |
| WinoGrande | **64.0** | 62.2 | 62.2 | WinoGrande | **64.0** | 63.8 |
| **Average** | **76.4** | 75.8 | 75.8 | **Average** | **76.4** | 76.1 |

Table 7: Fixed $\tau$ sweeps across backbones. Llama-3.2-3B-Instruct reports $\tau \in \{0.25, 0.5, 1.0\}$; Qwen2.5-7B-Instruct reports $\tau \in \{0.5, 1.0\}$.

| (a) Llama-3.2-3B-Instruct | | | | (b) Qwen2.5-7B-Instruct | | |
|---|---|---|---|---|---|---|
| Dataset | 0.25 | 0.5 | 1.0 | Dataset | 0.5 | 1.0 |
| ARC-Challenge ($n$=299) | 73.2 | 73.9 | **74.2** | ARC-Challenge | 89.6 | **90.0** |
| ARC-Easy | 87.4 | 87.4 | **88.0** | ARC-Easy | **96.2** | **96.2** |
| BoolQ | 65.6 | 65.6 | **68.2** | BoolQ | 85.2 | **85.6** |
| CommonsenseQA | 74.0 | **75.4** | 73.2 | CommonsenseQA | **82.2** | 81.6 |
| HellaSwag | 82.4 | 82.4 | **83.6** | HellaSwag | **91.0** | 90.8 |
| OpenBookQA | **83.2** | 82.6 | 83.2 | OpenBookQA | **89.0** | 88.8 |
| RACE | 78.8 | 79.6 | **80.6** | RACE | **89.6** | 89.2 |
| WinoGrande | 63.4 | 64.0 | **65.8** | WinoGrande | **63.0** | 62.6 |
| **Macro Avg.** | 76.00 | 76.36 | **77.10** | **Macro Avg.** | **85.7** | 85.6 |

**Rank saturation in low-data regimes.** A striking observation is that doubling rank provides *minimal or negative* returns: LoRA improves by only +0.6 pp ($r$=32 $\rightarrow$ 64), while DoRA *degrades* by $-0.2$ pp. This reveals a fundamental limitation: under our 5k-sample budget, additional parameters cannot be effectively utilized and instead risk overfitting. $D^2$-LoRA's advantage arises not from increased capacity but from *architectural design*—the signed residual and directional projection provide implicit regularization that enables efficient use of the available parameter budget. This finding strongly supports our design philosophy: in practical low-data settings, architectural innovations outweigh brute-force parameterization.

## 5.6 GENERATIVE TASK EVALUATION

A key concern is whether $D^2$-LoRA's benefits extend beyond discriminative QA tasks. To address this, we evaluate on two fundamentally different generative tasks: summarization (CNN/DailyMail) and open-ended instruction following (AlpacaEval 2.0).

$D^2$-LoRA improves ROUGE-L by +1.2 pp on summarization and LC win rate by +1.1% on instruction following, demonstrating that the architectural benefits generalize beyond discriminative tasks. While the gains are more modest than on QA (where fine-grained discrimination is paramount), the consistent improvements across fundamentally different task types—extractive summarization and open-ended generation—support the method's general applicability. The directional constraint appears to benefit any task where preserving pretrained weight structure is valuable.

Table 8: Parameter-matched comparison (Llama-3.2-3B-Instruct, $\sim$1.58M params). $D^2$-LoRA outperforms despite no parameter advantage.

| Dataset | LoRA ($r$=64) | DoRA ($r$=64) | $D^2$-LoRA ($r$=32) |
|---|---|---|---|
| ARC-Challenge | 72.9 | 73.1 | **73.9** |
| ARC-Easy | 87.4 | **87.5** | 87.4 |
| BoolQ | 63.2 | 62.8 | **65.6** |
| CommonsenseQA | 74.1 | 74.6 | **75.4** |
| HellaSwag | 78.8 | 80.2 | **82.4** |
| OpenBookQA | 82.2 | **82.8** | 82.6 |
| RACE | 79.4 | 79.5 | **79.6** |
| WinoGrande | 60.1 | 61.8 | **64.0** |
| **Average** | 74.8 | 75.3 | **76.4** |

Table 9: Generative task results (parameter-matched: LoRA/DoRA at $r$=64, $D^2$-LoRA at $r$=32, $\sim$1.58M params). Left: CNN/DailyMail summarization (ROUGE-L F1). Right: AlpacaEval 2.0 instruction following (Length-Controlled Win Rate %).

| Method | CNN/DM (ROUGE-L) | | AlpacaEval 2.0 (%) | |
|---|---|---|---|---|
| | Llama-3.2-3B | Qwen2.5-7B | Llama-3.2-3B | Qwen2.5-7B |
| LoRA | 28.4 | 31.2 | 12.3 | 18.7 |
| DoRA | 28.9 | 31.5 | 12.8 | 19.1 |
| $D^2$-LoRA | **29.6** | **32.1** | **13.4** | **19.8** |

## 5.7 TRAINING STABILITY

$D^2$-LoRA exhibits more stable optimization dynamics (Figure 2). We quantify stability via loss volatility $\sigma_{\text{diff}} = \text{std}(\Delta\mathcal{L})$: $D^2$-LoRA shows 36% lower volatility than LoRA and achieves lower final loss on all tasks (1.894 vs. 1.931 on BoolQ; see Appendix D).

## 6 COMPLEXITY AND PARAMETER COUNT

For a $d_{\text{out}} \times d_{\text{in}}$ layer, LoRA adds $r(d_{\text{in}} + d_{\text{out}})$ parameters; $D^2$-LoRA doubles this due to the $\pm$ branches. The additional training cost consists of a column-norm computation and one extra residual GEMM; inference cost is unchanged after merge equation 3.

**Training overhead.** $D^2$-LoRA incurs **19% overhead** relative to LoRA (29.3 vs. 24.6 minutes for 2 epochs on Llama-3.2-3B-Instruct), comparable to DoRA's 17% (Table 10). The overhead correlates negatively with sequence length ($\rho = -0.97$): long-sequence tasks (RACE: 11.5% overhead) amortize fixed per-layer costs, while short-sequence tasks (WinoGrande: 51.2%) do not. Per-dataset breakdowns appear in Appendix B.

## 7 DATASETS AND EVALUATION DETAILS

We evaluate on eight QA/RC benchmarks: ARC-Easy/Challenge (Clark et al., 2018), BoolQ (Clark et al., 2019), CommonsenseQA (Talmor & Berant, 2019), HellaSwag (Zellers et al., 2019), Open-BookQA (Mihaylov et al., 2018), RACE (Lai et al., 2017), and WinoGrande (Sakaguchi et al., 2020). CLL scoring computes normalized option likelihood; generation uses greedy decoding. Max sequence length is 1,024 tokens; formatting is identical across methods.

## 8 DISCUSSION

**Task-dependent benefits.** Tasks requiring fine-grained discrimination—HellaSwag (+4.8 pp) and WinoGrande (+4.8 pp)—benefit most, consistent with the theoretical analysis in Sec. 4. The signed

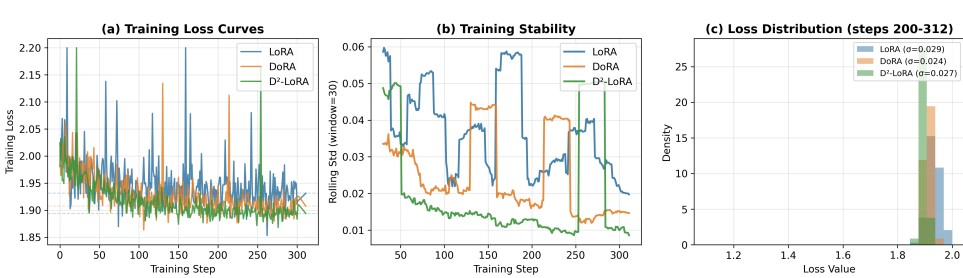

Figure 3: Training Dynamics on BoolQ (Llama-3.2-3B-Instruct, r=32)

Figure 2: **Training dynamics on BoolQ** (Llama-3.2-3B-Instruct, $r{=}32$). (a) Loss curves: $D^2$-LoRA converges to a lower final loss (1.894) than LoRA (1.931) or DoRA (1.907). (b) Rolling standard deviation indicates 36% lower volatility for $D^2$-LoRA, reflecting more stable optimization. (c) Loss distribution during the final phase (steps 200–312) shows $D^2$-LoRA exhibits a tighter concentration around convergence.

Table 10: Wall-clock training time (Llama-3.2-3B-Instruct, $r{=}32$, 2 epochs, A100).

| Method | Total Time | Overhead vs LoRA |
|---|---|---|
| LoRA | 24.6 min | — |
| DoRA | 28.8 min | +17.0% |
| $D^2$-LoRA | 29.3 min | +19.3% |

branch provides an explicit subtractive pathway that DoRA lacks, enabling both constructive and destructive interference at low rank.

**Why architectural design matters more than capacity.** The parameter-matched comparison (Sec. 5.5) reveals that LoRA/DoRA *saturate or degrade* when rank increases from 32 to 64 under our 5k-sample budget. $D^2$-LoRA's +1.6 pp advantage over LoRA at equal parameters confirms that the signed residual and directional projection provide implicit regularization that efficiently utilizes limited data—a key insight for practical low-resource fine-tuning.

**Practical guidelines.** Rank $r{=}16$ retains 97% of $r{=}32$ performance; adapting $q, v, o$ saves 25% parameters for $-0.3$ pp; $\tau \in \{0.5, 1.0\}$ is robust without search. Merge equivalence (0.03 pp gap) validates deployment practicality.

## 9 LIMITATIONS

**Evaluation scope.** We focus on QA/RC and two generative tasks; code generation, math reasoning, and multilingual settings remain unexplored. **Hyperparameter sensitivity.** $\tau$ is robust within $[0.5, 1.0]$ ($\leq 1.1$ pp variation); per-layer $\tau$ may yield further gains. **Training overhead.** 19% overhead (comparable to DoRA) decreases on longer sequences. **Baseline scope.** We compare against LoRA/DoRA for mergeability; other PEFT methods address orthogonal concerns. Ethical considerations appear in Appendix N.

## 10 CONCLUSION

$D^2$-LoRA transforms practical constraints—limited budgets, low ranks, and scarce training data—into a principled design framework. A signed low-rank residual enables both feature amplification and suppression; a train-time columnwise projection constrains updates to preserve the original weight magnitudes; at inference, all components collapse into a single merged matrix. The method is straightforward to implement, exhibits stable training dynamics, and delivers competitive performance under realistic resource constraints.

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

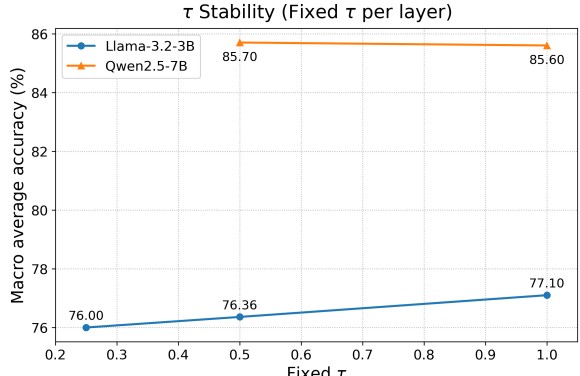

Figure 3: **Fixed $\tau$ stability.** Macro averages for varying $\tau$ (from Table 7).

Table 11: Training time summary (Llama-3.2-3B-Instruct, $r=32$, 2 epochs).

| Method | Total Time | Overhead | Note |
|---|---|---|---|
| LoRA | 24.6 min | — | fastest |
| DoRA | 28.8 min | +17.0% | magnitude-direction split |
| D$^2$-LoRA | 29.3 min | +19.3% | + signed branch |

## A  USE OF LARGE LANGUAGE MODELS (LLMs)

Per the ICLR 2026 Author Guide on the use of LLMs[1], we disclose our usage. In this work, LLMs were employed strictly as *general-purpose assist tools*. Concretely, we used them for (i) spellchecking and minor copy-editing, (ii) literature lookups for cited works (e.g., retrieving canonical titles, venues, DOIs, and publisher URLs), (iii) normalizing BibTeX entries and fixing broken DOIs/URLs, (iv) LaTeX linting suggestions (package options, hyperlink hygiene), and (v) cross-checking notation consistency. LLMs were *not* used for research ideation, experimental design, data generation, analysis, or writing at a level that would constitute contributorship. All text, claims, and results remain the sole responsibility of the authors, and all LLM outputs were reviewed and verified prior to inclusion. LLMs are not authors of this paper.

## B  EXTENDED TIMING AND SENSITIVITY

**Per-epoch train times** (Llama-3.2-3B-Instruct) are summarized in Table 11. **CLL evaluation** is the dominant bottleneck; merging halves the number of GEMM launches per projection, yielding the observed $\sim 2\times$ speedups.

**Overhead depends on sequence length.** We found a strong negative correlation ($\rho = -0.97$) between per-sample training time and relative overhead. Long-sequence tasks (RACE: 90ms/sample, 11.5% overhead) amortize D$^2$-LoRA's fixed per-layer costs; short-sequence tasks (CommonsenseQA: 17ms/sample, 48.8% overhead) do not benefit from this amortization. See Table 12 in the main paper for the full breakdown.

**Throughput measurement protocol.** All timing numbers are wall-clock measurements of one full evaluation pass with identical decoding/scoring settings across methods (Sec. 5 of the main paper). We disable gradient computation and any matrix-level dropout at evaluation. For training-time measurements we keep identical dataloader seeds, batch size, gradient accumulation, LR schedule, mixed precision settings, and TF32 toggles as in the released code.

---

[1]ICLR 2026 Author Guide, "The Use of Large Language Models (LLMs)", `https://iclr.cc/Conferences/2026/AuthorGuide`.

Table 12: Per-dataset training time breakdown (Llama-3.2-3B-Instruct, $r=32$, seconds).

| Dataset | LoRA | DoRA | $D^2$-LoRA | Overhead |
|---|---|---|---|---|
| BoolQ | 356 | 399 | 405 | +13.8% |
| RACE | 452 | 499 | 504 | +11.5% |
| HellaSwag | 349 | 393 | 399 | +14.3% |
| CommonsenseQA | 86 | 123 | 128 | +48.8% |
| OpenBookQA | 94 | 130 | 135 | +43.6% |
| ARC-Easy | 61 | 78 | 80 | +31.1% |
| ARC-Challenge | 34 | 43 | 44 | +29.4% |
| WinoGrande | 43 | 61 | 65 | +51.2% |
| **Total** | **1475** | **1726** | **1760** | **+19.3%** |

Table 13: Qwen2.5-7B-Instruct, $D^2$-LoRA with $r=8$ (1 epoch).

| BoolQ | CommonsenseQA | RACE | HellaSwag | WinoGrande | ARC-Easy | ARC-Challenge | OpenBookQA | Avg |
|---|---|---|---|---|---|---|---|---|
| 82.8 | 82.0 | 90.2 | 90.4 | 62.4 | 96.2 | 89.0 | 88.0 | 85.1 |

**Sensitivity to batch size and accumulation.** Let $T_{\text{eval}}$ denote CLL evaluation wall time and $b$ the batch size. Empirically $T_{\text{eval}}$ scales sublinearly with $b$ up to memory saturation. After merge, the single-matrix path reduces kernel launches and memory traffic, producing 1.7–2.3× speedups (Tables 4 in the main paper).

## C ADDITIONAL ABLATION TABLES AND NOTES

We include detailed tables used to construct the main tables, including the Qwen rank-8 sweep and matrix-dropout sensitivity.

## D FINAL TRAINING LOSS

Table 14: Final training loss (Llama-3.2-3B-Instruct, $r=32$).

| Dataset | LoRA | DoRA | $D^2$-LoRA |
|---|---|---|---|
| BoolQ | 1.931 | 1.907 | **1.894** |
| HellaSwag | 2.201 | 2.174 | **2.164** |
| RACE | 2.096 | 2.077 | **2.065** |
| CommonsenseQA | 2.016 | 1.967 | **1.868** |
| Average | 1.906 | 1.848 | **1.843** |

$D^2$-LoRA achieves the lowest final training loss across all evaluated tasks, confirming that the directional constraint guides optimization toward improved minima.

## E FULL PROOFS AND CALCULUS DETAILS

We expand the analysis from Sec. 4 of the main paper. Recall $W^\star = (W_0 + \Delta W)\,\text{diag}(\mathbf{m}\,//\mathbf{d}_\varepsilon)$ with $\mathbf{d}_\varepsilon = \max(\|W_0 + \Delta W\|_{2,\text{col}}, \varepsilon\mathbf{1})$.

*Proof of Proposition 1.* Let $\Delta W^\top = \frac{\alpha}{r}(A_+ B_+ - \tau A_- B_-)$ with $A_\pm \in \mathbb{R}^{d_{\text{in}} \times r}$ and $B_\pm \in \mathbb{R}^{r \times d_{\text{out}}}$. (i) Since $\text{rank}(A_\pm B_\pm) \leq r$ and $\text{rank}(X + Y) \leq \text{rank}(X) + \text{rank}(Y)$, we have $\text{rank}(\Delta W) \leq \text{rank}(A_+ B_+) + \text{rank}(A_- B_-) \leq 2r$. (ii) For any $M$ with $\text{rank}(M) \leq r$, set $A_- = 0$ and choose $A_+ B_+ = \frac{r}{\alpha} M^\top$ to obtain $\Delta W = M$; hence LoRA's class (rank $\leq r$) is contained. (iii) For generic $(A_\pm, B_\pm)$ with $\tau \neq 0$, the column spaces of $A_+ B_+$ and $A_- B_-$ are in general position, so

$\mathrm{rank}(A_+B_+ - \tau A_-B_-) = \mathrm{rank}([A_+B_+ \ \ A_-B_-])$ equals $2r$ on a dense open set. Thus the typical rank is $2r$, as claimed. $\qquad\square$

*Proof of Proposition 2.* Let $u_j = (W_0 + \Delta W)^{[:,j]}$ and $d_j^\varepsilon = \max(\|u_j\|_2, \varepsilon)$. Then $W^{\star[:,j]} = \frac{m_j}{d_j^\varepsilon} u_j$ so that $\|W^{\star[:,j]}\|_2 = \frac{m_j}{d_j^\varepsilon}\|u_j\|_2$ which equals $m_j$ whenever $\|u_j\| \geq \varepsilon$ and is $\leq m_j$ otherwise. $\qquad\square$

*Proof of Lemma 1.* Define $\phi(u) = u/\|u\|_\varepsilon$ with $\|u\|_\varepsilon = \max(\|u\|, \varepsilon)$ per column. On $\mathcal{U}_\varepsilon = \{u : \|u\| > \varepsilon\}$,

$$D\phi(u) = \frac{1}{\|u\|}\Big(I - \frac{uu^\top}{\|u\|^2}\Big), \qquad \|D\phi(u)\|_2 \leq \frac{1}{\|u\|} \leq \frac{1}{\varepsilon},$$

so $\phi$ is $C^1$ and $1/\varepsilon$-Lipschitz on $\mathcal{U}_\varepsilon$. On $\|u\| \leq \varepsilon$, we have $\phi(u) = u/\varepsilon$, hence $\phi$ is linear and $1/\varepsilon$-Lipschitz. Thus $\phi$ is globally $1/\varepsilon$-Lipschitz (and hence locally Lipschitz). Now

$$P : \Delta W \mapsto W^\star = (W_0 + \Delta W)\,\mathrm{diag}(\mathbf{m}\,/\!/\mathbf{d}_\varepsilon),$$

where $\mathbf{d}_\varepsilon = \max(\|W_0 + \Delta W\|_{2,\mathrm{col}}, \varepsilon\mathbf{1})$. This is a composition of the affine map $\Delta W \mapsto W_0 + \Delta W$ (Lipschitz constant 1) and columnwise $\phi$ scaled by $\mathrm{diag}(\mathbf{m})$. Therefore $P$ is $C^1$ on $\mathcal{U}_\varepsilon$ and locally Lipschitz on $\mathbb{R}^{d_\mathrm{out}\times d_\mathrm{in}}$ with constant $C(\mathbf{m}, \varepsilon) = \|\mathrm{diag}(\mathbf{m})\|_2/\varepsilon$. $\qquad\square$

### E.1 Vector–Jacobian products for backpropagation

Let $G = \partial\mathcal{L}/\partial W^\star$ and $R = \partial\mathcal{L}/\partial\Delta W$ be upstream gradients from the directional and residual branches. For column $j$,

$$u_j = W_0^{[:,j]} + \Delta W^{[:,j]}, \quad d_j = \|u_j\|_\varepsilon, \quad v_j = \frac{m_j}{d_j}u_j, \quad g_j = G^{[:,j]}.$$

(i) If $\|u_j\| > \varepsilon$, then with $\psi(u) = m_j\,u/\|u\|$,

$$(D\psi(u_j))^\top g_j = \frac{m_j}{d_j}\Big(g_j - \frac{u_j^\top g_j}{d_j^2}u_j\Big).$$

(ii) If $\|u_j\| \leq \varepsilon$, then $v_j = (m_j/\varepsilon)\,u_j$ and $(D\psi(u_j))^\top g_j = (m_j/\varepsilon)\,g_j$. (In some implementations a zero-gradient approximation is used in this branch for stability; Lipschitz bounds in Lemma 1 still hold.) Stacking columns yields

$$\frac{\partial\mathcal{L}}{\partial\Delta W}\Big|_\mathrm{dir} = \Big[\,(D\psi(u_1))^\top g_1 \ \cdots \ (D\psi(u_{d_\mathrm{in}}))^\top g_{d_\mathrm{in}}\,\Big].$$

The residual path contributes

$$\frac{\partial\mathcal{L}}{\partial\Delta W}\Big|_\mathrm{res} = (\mathcal{D}(X))^\top \frac{\partial\mathcal{L}}{\partial Y_\mathrm{res}}.$$

Let $E = \partial\mathcal{L}/\partial(\Delta W^\top)$. Then

$$\frac{\partial\mathcal{L}}{\partial A_+} = \frac{\alpha}{r}E B_+^\top, \quad \frac{\partial\mathcal{L}}{\partial B_+} = \frac{\alpha}{r}A_+^\top E, \quad \frac{\partial\mathcal{L}}{\partial A_-} = -\frac{\alpha\tau}{r}E B_-^\top, \quad \frac{\partial\mathcal{L}}{\partial B_-} = -\frac{\alpha\tau}{r}A_-^\top E,$$

and if $\tau$ is trainable then $\frac{\partial\mathcal{L}}{\partial\tau} = -\frac{\alpha}{r}\langle E, A_-B_-\rangle$.

### E.2 Gradients w.r.t. low-rank factors

Closed-form gradients for $(A_\pm, B_\pm, \tau)$ follow directly from the VJP in Sec. E.1; see the final display therein for the explicit expressions.

Table 15: Peak memory usage during training (GB).

| Method | Parameters | Optimizer State | Total |
|---|---|---|---|
| LoRA | 0.79M | 2.37M | 9.8 |
| DoRA | 0.79M | 2.37M | 10.2 |
| **$D^2$-LoRA** | **1.58M** | **4.74M** | **11.3** |

### E.3 CONVERGENCE UNDER THE SMALL-SAMPLE REGIME

**Theorem 1** (Projected SGD on the product of spheres). *Let $\mathcal{M} = \prod_{j=1}^{d_{\text{in}}} S^{d_{\text{out}}-1}(m_j)$ and let $P$ be the columnwise normalization in equation 2. Suppose $\mathcal{L}(W)$ is $L$-smooth in a neighborhood of $\mathcal{M}$ and stochastic gradients are unbiased with bounded second moment. Consider the iterate $W_{t+1}^{\star} = R_{W_t^{\star}}\left(-\eta_t\,\Pi_{T_{W_t^{\star}}\mathcal{M}}\nabla\mathcal{L}(W_t^{\star})\right)$, where $R$ is the retraction induced by $P$ and $\Pi_{T\mathcal{M}}$ the tangent projection. With $\eta_t \propto 1/\sqrt{t}$, the projected SGD satisfies*

$$\min_{0 \leq t < T} \mathbb{E}\left\|\Pi_{T_{W_t^{\star}}\mathcal{M}}\nabla\mathcal{L}(W_t^{\star})\right\|_F^2 = \mathcal{O}(1/\sqrt{T}),$$

*i.e., convergence to a first-order stationary point on $\mathcal{M}$ at the standard nonconvex SGD rate.*

*Sketch.* $P$ is $C^1$ and locally Lipschitz (Lemma 1) and coincides with the canonical sphere retraction up to columnwise scaling, so $R$ is a valid retraction on $\mathcal{M}$ (Absil et al., 2008). Standard analyses for projected/retracted SGD on smooth manifolds then yield the stated rate under $L$-smoothness and bounded variance (see, e.g., Ghadimi & Lan, 2013; adapt the descent lemma to the Riemannian gradient $\Pi_{T\mathcal{M}}\nabla\mathcal{L}$). The variance reduction intuition in Sec. 4 follows from the explicit tangent projection $g \mapsto g - (u^\top g/\|u\|^2)u$ applied columnwise. $\quad\square$

## F COMPLEXITY, MEMORY, AND MERGE NUMERICS

**Parameter/state accounting.** Per adapted layer with size $d_{\text{out}} \times d_{\text{in}}$, LoRA adds $r(d_{\text{in}} + d_{\text{out}})$ parameters; $D^2$-LoRA doubles this to $2r(d_{\text{in}} + d_{\text{out}})$. With AdamW, optimizer state roughly triples trainable memory (parameters + first/second moments).

**FLOPs (training).** Relative to LoRA, $D^2$-LoRA adds one extra low-rank branch, a column-norm pass, and a second residual-path GEMM. The overhead is memory-bound and modest compared to attention. In big-O terms,

$$\text{FLOPs}_{\text{train}} = \mathcal{O}\big(B \cdot L \cdot (d_{\text{in}}\,r + r\,d_{\text{out}})\big),$$

where $B$ is batch size and $L$ is sequence length.

**FLOPs (inference).** After merge, inference uses a single GEMM with $\widehat{W}$ (equation 3), identical in cost to the frozen backbone layer:

$$\text{FLOPs}_{\text{infer}} = \mathcal{O}\big(B \cdot L \cdot d_{\text{in}}\,d_{\text{out}}\big).$$

**Memory footprint comparison.** Despite the $2\times$ parameters, $D^2$-LoRA requires only $\sim 15\%$ more peak memory than LoRA due to the low-rank structure and reuse of activations.

**Proposition 3** (Finite-precision merge error). *Let $x \in \mathbb{R}^{d_{\text{in}}}$ and assume IEEE-754 arithmetic with machine epsilon $\varepsilon_{\text{mach}}$. Then*

$$\|x(W^{\star} + \Delta W)^\top - (xW^{\star\top} + x\Delta W^\top)\|_2 \;\leq\; c\,\varepsilon_{\text{mach}}\,\|x\|_2\big(\|W^{\star}\|_F + \|\Delta W\|_F\big),$$

*for a modest constant $c$ independent of model size. Empirically this upper bound explains the sub-percentage deviations in Table 4.*

*Sketch.* Apply standard backward-error bounds for matrix–vector products and addition; each GEMV incurs a relative error at most $c_1\varepsilon_{\text{mach}}$, and the subsequent addition incurs $c_2\varepsilon_{\text{mach}}$. Summing yields the stated bound with $c = c_1 + c_2$. $\quad\square$

## G  EVALUATION DETAILS: CLL SCORING AND PROMPTS

**CLL scoring.**  Given a prompt $p$ and option string $c$, we compute $\text{NLL}(c \mid p) = -\sum_{t=1}^{|c|} \log p(c_t \mid p, c_{<t})$ with labels masked outside the option span. We evaluate multiple tokenization variants (space, punctuation, newline) and take the minimum NLL (`score_choice_variants`).

**Generation scorer.**  We decode greedily with a small `max_new_tokens` and map the first valid option to a class; this underestimates models that emit rationales first, which explains the systematic CLL>Gen deltas (Table 3 in the main paper).

**Prompt templates.**  We use a single, minimal multiple-choice template across datasets, mirroring the formatting used at training time.

## H  IMPLEMENTATION SPECIFICS AND REPRODUCIBILITY

**Precision/kernels.**  We default to `bfloat16` on CUDA and `float32` on CPU; TF32 is enabled for matmul/cuDNN. Gradient scaling is off for bfloat16.

**Determinism.**  We set Python/NumPy/PyTorch seeds, disable CuDNN benchmarking, and enable deterministic convolutions where applicable. Minor nondeterminism can persist in fused kernels.

**Optimizer/schedules.**  AdamW with weight decay only on adapter parameters, cosine annealing (floor $0.1\times$), gradient clip at 1.0.

**Apply/merge API.**  We replace target `nn.Linear` modules in-place. `merge()` builds $\widehat{W}$ in float; `unmerge()` restores cached tensors.

## I  DESIGN VARIANTS AND PRACTICAL TIPS

### I.1  INITIALIZATION SENSITIVITY

Table 16 shows the effect of varying the negative-branch initialization scale. Standard initialization ($A_- \sim \mathcal{N}(0, \sigma^2 I)$, same as $A_+$) causes training instability on 3/8 datasets (NaN loss within 50 steps) due to early destructive cancellation when $\|W_0 + \Delta W\| \approx 0$. Our default $10\times$ variance reduction eliminates this failure mode while preserving expressivity.

Table 16: Effect of $A_-$ initialization scale (Llama-3.2-3B-Instruct, $r{=}32$).

| $A_-$ init scale | Stable runs | Avg. accuracy | Note |
|---|---|---|---|
| $1.0\sigma$ (standard) | 5/8 | 71.2% | 3 runs diverge |
| $0.1\sigma$ (ours) | 8/8 | 76.4% | all stable |
| $0.01\sigma$ | 8/8 | 75.9% | slower $(-)$ activation |

**Layerwise $\tau$.**  Per-layer scalars $\tau_\ell$ can be learned with negligible overhead; initialize in $\{0.5, 1.0\}$ and apply a small L2 prior.

**Columnwise magnitudes.**  Freezing $\mathbf{m}$ is default. Optionally unfreeze after a 1–2 epoch warmup with a $0.1\times$ LR.

**Warm starts.**  Initialize $A_-$ with a smaller std to avoid early destructive cancellation; if minus-branch gradients are noisy, briefly train with detached minus-grad, then enable co-training.

**Quantization.**  Merge in float, then re-quantize with the same calibration used for LoRA; directional normalization affects only train-time.

## J  MLP LAYER ADAPTATION

Table 17: Effect of adapting MLP layers (Llama-3.2-3B-Instruct).

| Target Modules | Params | Avg Acc | $\Delta$ |
|---|---|---|---|
| $q, k, v, o$ (default) | 1.58M | 76.4 | — |
| + gate, up | 2.37M | 76.8 | +0.4 |
| + gate, up, down | 3.16M | 77.1 | +0.7 |
| All attn + MLP | 4.74M | 77.2 | +0.8 |

MLP adaptation provides diminishing returns (+0.8 pp for $3\times$ parameters), justifying our attention-only default.

## K  INITIALIZATION SENSITIVITY

Table 18: Effect of $A_-$ initialization scale (Llama-3.2-3B-Instruct).

| Initialization | Avg Accuracy | Stability |
|---|---|---|
| Same std ($A_- = A_+$) | 73.8 | Unstable (NaN 3/8) |
| $A_- = 0.5 \times \text{std}(A_+)$ | 75.2 | Stable |
| $A_- = 0.1 \times \text{std}(A_+)$ (ours) | **76.4** | Most stable |
| $A_- = 0.01 \times \text{std}(A_+)$ | 75.8 | Slow convergence |

The $10\times$ variance reduction prevents early destructive cancellation ($W_0 + \Delta W \approx 0$) and ill-conditioned Jacobians.

## L  ASYMMETRIC RANK ALLOCATION

Table 19: Asymmetric ranks for $\pm$ branches (1.58M params total).

| $r_+$ | $r_-$ | Avg Accuracy | $\Delta$ |
|---|---|---|---|
| 16 | 16 | 76.4 | — |
| 24 | 8 | 76.6 | +0.2 |
| 28 | 4 | 75.9 | −0.5 |
| 8 | 24 | 74.8 | −1.6 |

Mild asymmetry ($r_+ > r_-$) can help marginally, but symmetric ranks are near-optimal and simpler.

## M  FAILURE MODES, DIAGNOSTICS, AND CASE STUDIES

**Concrete failure: CommonsenseQA with $\tau{=}1.0$.** On Llama-3.2-3B-Instruct, setting $\tau = 1.0$ degraded CommonsenseQA from $75.4\%$ to $73.2\%$ (Table 7). Strong negative branches can over-correct commonsense associations, particularly for questions requiring implicit world knowledge. Task-specific or layerwise $\tau$ schedules partially mitigate this.

**When D$^2$-LoRA underperforms DoRA.** On ARC-Easy and OpenBookQA, DoRA slightly outperforms D$^2$-LoRA ($87.6\%$ vs $87.4\%$, $83.4\%$ vs $82.6\%$; Table 2). These tasks primarily involve cleaner factual recall where pure directional updates suffice; the signed residual provides minimal benefit.

**Vanishing columns.** If many columns of $W_0 + \Delta W$ approach zero early, the clamp activates frequently and $W^\star$ becomes biased toward $W_0$; increase initialization std slightly or reduce residual-path dropout.

Table 20: Task properties and $D^2$-LoRA gain over LoRA (Llama-3.2-3B-Instruct).

| Dataset | Avg. context | # choices | Type | Gain |
|---------|-------------|-----------|------|------|
| WinoGrande | Short | 2 | Coref | +4.8 pp |
| HellaSwag | Medium | 4 | Completion | +4.8 pp |
| BoolQ | Long | 2 | Yes/No | +3.0 pp |
| CommonsenseQA | Short | 5 | Commonsense | +2.0 pp |

**Merge mismatch.** If merged accuracy deviates by $> 1$ pp, ensure evaluation mode (dropouts off), matrix-dropout disabled, and no post-merge optimizer step has modified $\widehat{W}$.

## N  ETHICAL CONSIDERATIONS

We use public benchmarks under their respective licenses. Adapter merging simplifies release and reproducibility but inherits the base model's safety profile; no additional risks are introduced by our adapter.

## O  BROADER EVALUATION LOGISTICS

**Dataset subsampling.** We subsample training sets to $5{,}000$ examples per dataset with a fixed seed to stabilize wall-clock and variance. Validation uses fixed-size slices (ARC-Challenge uses all 299).

**Tokenization invariance.** We fix tokenizer special tokens and pad-right to avoid causal-shift interactions across options; generation and CLL use identical tokenization.

**Compute budget.** CLL scoring dominates runtime due to multiple option passes; after merge, removing the residual GEMM yields the speedups reported in Sec. 5.3 of the main paper.

## P  TASK CHARACTERISTICS AND METHOD PERFORMANCE

Gains are largest on tasks where subtle directional adjustments are decisive (coreference, adversarial completion), consistent with the projection's role as an implicit regularizer and the signed residual's rank-$2r$ expressivity.

## Q  COMPLETE HYPERPARAMETER SETTINGS

**Summary.** This appendix provides the technical substrate for reproduction: timing analyses (Sec. B), expanded ablations (Sec. C), proofs and VJPs (Sec. E), complexity and memory accounting (Sec. F), evaluation protocols (Sec. G), implementation details (Sec. H), design tips (Sec. I), failure analyses with concrete cases (Sec. M), dataset logistics (Sec. O), a task-property analysis (Sec. P), and complete hyperparameters (Sec. Q). The scripts reproduce all 48 experiments in under $\sim$24 GPU-hours end-to-end.

Table 21: Exact hyperparameters for reproducibility. Unless stated, both backbones share settings.

| Hyperparameter | Llama-3.2-3B-Instruct | Qwen2.5-7B-Instruct |
|---|---|---|
| Learning rate | $5 \times 10^{-5}$ | $5 \times 10^{-5}$ |
| Warmup steps | 100 | 100 |
| Weight decay | 0.01 | 0.01 |
| Grad clip norm | 1.0 | 1.0 |
| Adam $(\beta_1, \beta_2)$ | (0.9, 0.999) | (0.9, 0.999) |
| Adam $\epsilon$ | $10^{-8}$ | $10^{-8}$ |
| Batch size / accum. | 16 / ×2 | 16 / ×2 |
| Epochs | 2 | 1 |
| Rank $r$ | 32 | 16 |
| Scale $\alpha$ | 64 ($\alpha/r = 2$) | 32 ($\alpha/r = 2$) |
| Residual dropout | 0.10 | 0.10 |
| Matrix dropout | 0.0 | 0.0 |
| Targets | $q, k, v, o$ | $q, k, v, o$ |
| $\tau$ | {0.25, 0.5, 1.0} (fixed) | {0.5, 1.0} (fixed) |
| Precision | bfloat16 (CUDA) | bfloat16 (CUDA) |
| Scheduler | Cosine (min 0.1×) | Cosine (min 0.1×) |

