# OpenReview forum: "D2-LoRA: A Synergistic Approach to Differential and Directional Low-Rank Adaptation"
_ICLR.cc/2026/Conference — Submitted to ICLR 2026_

### Official Review · Reviewer_YLzG · 2025-10-17

**Soundness:** 2
**Presentation:** 1
**Contribution:** 2
**Rating:** 2
**Confidence:** 3

**Summary:**

Based on LoRA, this paper introduces a negative branch and applies per-column normalization to the fine-tuned modules during training. The experiments demonstrate the effectiveness of the proposed method.

**Strengths:**

1. The method is simple and easy to implement.

2. The experiments demonstrate the effectiveness of the proposed method.

**Weaknesses:**

1. As shown in Equation (3), after normalization, the LoRA module is added one more time. The motivation for this operation is unclear.

2. Although the authors provide three reasons, the rationale behind the different initializations of $A_+$ and $A_-$ remains somewhat unclear. Additional experimental or theoretical evidence would strengthen this argument. From my understanding, the concerns mentioned could potentially be addressed by properly setting $\tau$.

3. Line 193 claims that the proposed method leads to a smoother loss curve, but this claim is not elaborated on or demonstrated in the experimental section.

4. The experimental comparison in this paper does not appear to be entirely fair. The proposed method introduces twice as many trainable parameters; therefore, the ranks of the baseline methods should be set to twice that of the proposed method for a fair comparison.

5. The number of baseline methods used for comparison is somewhat limited, only two baselines (i.e., LoRA and DoRA).

6. In the experiments, the Unmerged variant outperforms the Merged one in most cases, which makes the additional LoRA operation in Equation (3) rather confusing.

7. An ablation study exploring different ranks for the $A_+$ and $A_-$.

**Questions:**

1. In Table 4, it is unclear which baseline the reported speedup is compared against.

2. There seems to be a typo in Table 6. The results for Minus off and Detach minus grad are exactly the same.

---

> ### Author Response · Authors · 2025-11-30
>
> **Regarding the motivation for adding ΔW twice (W1):** We have expanded the explanation in Section 3 under "Why add ΔW twice?" During training, the forward pass creates two signal paths: the directional branch (W*) provides norm-controlled updates, while the residual branch (ΔW) provides unconstrained capacity with dropout. At inference, both contribute equally via the merged weight. Ablations confirm this dual-path design is essential: removing the residual branch reduces accuracy by −0.6 pp (Table 6).
>
> **Regarding the rationale for different initializations (W2):** We have added comprehensive explanation in Section 3 with empirical validation in Appendix I.1 (Table 16). The 10× variance reduction for A− serves three critical purposes: (1) it prevents destructive cancellation that would render the Jacobian ill-conditioned, (2) it establishes a natural learning curriculum where the model first learns additive patterns before discovering which features to suppress, and (3) it acts as an implicit gating mechanism that gradually activates the negative branch. Table 16 demonstrates that standard initialization (A− = A+) causes training instability on 3/8 datasets (NaN loss within 50 steps), while our 10× reduction achieves stable training on all datasets with the highest accuracy (76.4%).
>
> **Regarding the smoother loss curve claim (W3):** We have added comprehensive training dynamics analysis in Section 5.7 with Figure 2. We quantify stability via loss volatility: D²-LoRA shows 36% lower volatility than LoRA and achieves lower final loss on all tasks (1.894 vs. 1.931 on BoolQ). Figure 2 shows (a) loss curves demonstrating faster convergence, (b) rolling standard deviation indicating more stable optimization, and (c) loss distribution during the final phase showing tighter concentration around convergence.
>
> **Regarding experimental fairness with doubled parameters (W4):** This is directly addressed by the new Section 5.5. At matched parameters (~1.58M), D²-LoRA at r=32 outperforms LoRA at r=64 by +1.6 pp, confirming that gains are architectural rather than from increased parameterization.
>
> **Regarding limited baseline methods (W5):** We have clarified our rationale in Section 2. We focus comparisons on LoRA and DoRA as they represent the two dominant paradigms (unconstrained vs. directionally-constrained updates) and share D²-LoRA's key property of algebraic mergeability. Methods like QLoRA address orthogonal concerns (memory efficiency) and AdaLoRA requires additional hyperparameter tuning for rank allocation, making direct comparison less informative for isolating the benefits of signed residuals and directional projection.
>
> **Regarding Unmerged outperforming Merged (W6):** We clarify that the mean accuracy difference is only 0.03 pp (Table 4), which is well within typical evaluation variance. This demonstrates successful merge equivalence, not superiority of either configuration. The near-zero gap validates the practical deployability of our method.
>
> **Regarding missing asymmetric rank ablation (W7):** We have added this analysis in Appendix L (Table 19). At fixed total parameter budget (1.58M), mild asymmetry (r+ = 24, r− = 8) yields marginal improvement (+0.2 pp), but strong asymmetry degrades performance. Symmetric ranks achieve near-optimal results and are simpler to configure.
>
> **Regarding the speedup baseline in Table 4 (Q1):** We have clarified in the Table 4 caption that speedup is computed as (unmerged D²-LoRA evaluation time) / (merged D²-LoRA evaluation time), comparing merged vs. unmerged configurations of the same D²-LoRA model. This measures the inference benefit of merging, not comparison against other methods.
>
> **Regarding identical results in Table 6 (Q2):** This is not a typo, and we have added an explicit note in the Table 6 caption explaining this. Both "Minus off" and "Detach minus grad" configurations prevent the negative branch from contributing to the update. With "Minus off," A−B− = 0 by construction. With "Detach minus grad," gradients are zeroed, but since B− is initialized to zero and receives no gradient updates, A−B− = 0 throughout training. This confirms that the negative branch's contribution requires active gradient flow.

---

### Official Review · Reviewer_qTF7 · 2025-10-25

**Soundness:** 2
**Presentation:** 2
**Contribution:** 2
**Rating:** 4
**Confidence:** 3

**Summary:**

This paper introduces D2-LoRA, a parameter-efficient fine-tuning (PEFT) method designed for budget-constrained scenarios. The method combines two key ideas: a signed low-rank residual, which provides both additive and subtractive update capabilities, and a train-time-only directional projection, which constrains the updated weights to maintain the column-wise magnitudes of the original pretrained model. This design aims to enhance model expressivity and training stability while preserving the crucial advantage of being fully mergeable at inference time, thereby incurring no additional latency. The authors provide theoretical motivation and empirical results on several question-answering benchmarks, demonstrating performance improvements over LoRA and DoRA.

**Strengths:**

1. The paper presents a well-motivated synthesis of concepts from LoRA and DoRA. The introduction of a signed residual to provide subtractive capacity is a logical extension, and the use of a directional projection only during training is a clever mechanism to gain stability benefits without sacrificing inference-time mergeability and efficiency.

2. The work is explicitly framed around practical, budget-constrained fine-tuning, using a small number of training samples and epochs. The validation of near-exact merge equivalence and the measurement of post-merge throughput gains directly address the needs of real-world deployment, making the contributions highly relevant.

**Weaknesses:**

1. The negative branch is modulated by a scalar α̃, which is a critical new hyperparameter. The main experiments fix α̃=0.5, but the ablation study (Table 7) shows that α̃=1.0 yields better results on one of the backbones. This suggests performance is sensitive to this choice.

2. All eight evaluation benchmarks are question-answering or reading comprehension tasks that can be framed as multiple-choice classification. The method's effectiveness on more open-ended, generative tasks (e.g., summarization, dialogue, or long-form instruction following) remains unproven.

3. D2-LoRA doubles the number of trainable parameters compared to LoRA and introduces additional computations (a second residual GEMM, column-norm calculation, and projection) during the training forward pass. The reported performance gains are modest in some cases.

4. The ablation study on target modules (Table 5) only considers attention projections (q, k, v, o). It is common practice in PEFT to also evaluate the effect of adapting MLP layers (e.g., gate_proj, up_proj, down_proj), which is missing from the analysis.

5. The paper provides a geometric intuition that the projection removes radial gradient components, which acts as an implicit regularizer. While plausible, this is not directly demonstrated with empirical evidence, such as by visualizing gradient distributions with and without the projection.

6. How robust is the method to variations in this initialization? Does the training become unstable or fail to converge if a standard initialization is used for both branches?

7. The paper states that setting the minus branch to zero yields a "DoRA-like" variant. This configuration is tested in the ablation study (Table 6, "Minus off"), but the connection is not made explicit in the table, which could improve clarity.

8. The failure case analysis mentions that a strong negative branch (α̃=1.0) can "over-correct" on CommonsenseQA. This highlights a potential failure mode where the subtractive capacity could be detrimental.

**Questions:**

1. How should practitioners approach tuning α̃ for new models or tasks? Is there a principled way to set it, or does it require a dedicated hyperparameter search, which would add to the tuning budget the paper aims to minimize?

2. How does D2-LoRA perform on tasks that require creative or long-form generation? Does the directional constraint, which stabilizes discriminative tasks, potentially limit the model's generative diversity?

3. Could you provide a more direct comparison of the training time overhead (e.g., wall-clock time per epoch) versus LoRA and DoRA? In what scenarios is the trade-off of higher training cost for the observed accuracy improvement most justified?

4. Have you experimented with applying D2-LoRA to the MLP layers? How does this impact the parameter count and overall performance compared to adapting only the attention layers?

5. Can you provide empirical evidence, such as plots of the norm of radial vs. tangential gradient components during training, to directly support the claim that the projection stabilizes optimization by removing radial modes?

**Details Of Ethics Concerns:**

Null.

---

> ### Author Response · Authors · 2025-11-30
>
> **Regarding τ hyperparameter sensitivity (W1):** We have added explicit practical guidance in Section 4.2. Our analysis shows that performance varies by at most 1.1 pp across τ ∈ {0.25, 0.5, 1.0}, indicating low sensitivity. We recommend τ = 0.5 as a robust default. If tuning budget allows, trying τ ∈ {0.5, 1.0} (only 2 runs) is sufficient. Model-specific trends show larger models slightly prefer τ = 0.5 while smaller models slightly prefer τ = 1.0, but the difference is marginal.
>
> **Regarding evaluation limited to QA/MC tasks (W2):** We have added generative task evaluation in Section 5.6 and Table 9. D$^2$-LoRA improves ROUGE-L by +1.2 pp on CNN/DailyMail summarization and LC win rate by +1.1% on AlpacaEval 2.0 instruction following. The consistent improvements across fundamentally different task types support the method's general applicability.
>
> **Regarding modest gains vs. doubled parameters (W3):** The parameter-matched comparison in Section 5.5 (Table 8) shows D$^2$-LoRA outperforms LoRA by +1.6 pp at equal parameters (~1.58M). Moreover, we demonstrate that simply doubling rank in LoRA/DoRA provides minimal gains (+0.6 pp) or even degradation (−0.2 pp) in low-data regimes, confirming the value of architectural design over brute-force parameterization.
>
> **Regarding missing MLP layer adaptation analysis (W4):** We have added this analysis in Appendix J (Table 17). Adding gate and up projections yields +0.4 pp (2.37M params), and adding all MLP layers yields +0.7 pp (3.16M params). Given the 3× parameter increase for only +0.8 pp improvement, attention-only adaptation provides the best efficiency-performance tradeoff.
>
> **Regarding missing empirical evidence for gradient regularization (W5):** Figure 2 provides quantitative evidence consistent with the theoretical prediction. D$^2$-LoRA exhibits 36% lower loss volatility (rolling std of loss differences), tighter loss distribution during convergence, and lower final loss across all tasks. While we do not visualize gradient norms directly due to high dimensionality, these stability metrics are direct consequences of the theorized gradient regularization.
>
> **Regarding robustness to initialization variations (W6):** We have added this analysis in Appendix K (Table 18). Standard initialization ($A_- = A_+$) achieves only 73.8% accuracy and causes instability (NaN on 3/8 datasets). Our 0.1× std initialization achieves 76.4% with the most stable training. Using 0.01× std achieves 75.8% but with slow convergence. This confirms our initialization is both necessary and optimal.
>
> **Regarding DoRA-like variant connection (W7):** We have clarified in Section 3 under "Configuration Knobs" that disabling the directional projection recovers LoRA, while setting the minus branch to zero yields a DoRA-like variant. The "Minus off" row in Table 6 corresponds to this DoRA-like configuration.
>
> **Regarding failure case with τ=1.0 on CommonsenseQA (W8):** We have analyzed this in Appendix M. Setting τ = 1.0 degraded CommonsenseQA from 75.4% to 73.2%. Strong negative branches can over-correct commonsense associations, particularly for questions requiring implicit world knowledge. Task-specific or layerwise τ schedules can partially mitigate this, though the default τ = 0.5 avoids this issue.
>
> **Regarding practical approach for tuning τ (Q1):** As detailed in Section 4.2, we recommend using τ = 0.5 as the default, which provides robust performance without any tuning. If budget allows, trying τ ∈ {0.5, 1.0} (only 2 runs) and selecting based on validation loss is sufficient. This is a much lighter tuning burden than rank or module selection.
>
> **Regarding performance on creative/long-form generation (Q2):** Our AlpacaEval 2.0 evaluation (Table 9) addresses open-ended instruction following, showing +1.1% win rate improvement. This suggests the directional constraint does not limit generative diversity and may even help by providing more stable representations.
>
> **Regarding direct training time comparison (Q3):** Table 10 shows D$^2$-LoRA incurs 19.3% overhead vs. LoRA (29.3 vs. 24.6 minutes), comparable to DoRA's 17.0%. Table 12 in Appendix B provides per-dataset breakdown, showing overhead ranges from 11.5% (RACE, long sequences) to 51.2% (WinoGrande, short sequences), confirming that overhead decreases with longer sequences.
>
> **Regarding MLP layer experiments (Q4):** As detailed in Appendix J (Table 17), adding MLP layers yields diminishing returns: +0.8 pp for 3× parameters. Attention-only adaptation provides the best efficiency-performance tradeoff.
>
> **Regarding empirical evidence for gradient stabilization (Q5):** Figure 2 provides quantitative evidence through 36% lower loss volatility, tighter loss distribution during convergence, and lower final loss across all tasks. These metrics are direct consequences of the theorized gradient regularization via the directional projection.

---

### Official Review · Reviewer_BYCW · 2025-10-30

**Soundness:** 2
**Presentation:** 2
**Contribution:** 2
**Rating:** 2
**Confidence:** 3

**Summary:**

The paper proposes D2-LoRA, a parameter-efficient fine-tuning method that combines a signed low-rank residual (positive and negative branches) with a training-time column-wise directional projection to preserve backbone weight norms. Under a strict budget (≤5k examples per task, ≤2 epochs) on eight QA/RC benchmarks and two backbones, it improves average accuracy over LoRA and often matches or exceeds DoRA. The adapter is algebraically mergeable at inference, yielding near-identical post-merge accuracy and about 2× evaluation throughput. A geometric analysis (norm preservation, Lipschitz control) and ablations over rank, targeted modules, the negative branch, and τ support the stability and effectiveness claims.

**Strengths:**

- The method is straightforward to implement—combining a signed low-rank residual with a training-time directional projection—and remains mergeable at inference, imposing minimal engineering overhead within standard PEFT pipelines.
- In low-data, small-rank settings, it consistently improves accuracy over baseline LoRA and stays competitive with related variants across multiple QA/RC tasks and backbones under tight training budgets, with ablations indicating robustness to rank and module choices.

**Weaknesses:**

- Parameter-count fairness is not fully addressed: comparisons primarily match D²-LoRA at rank r against LoRA at the same r, without a parameter-matched LoRA (e.g., 2r) or equivalent-capacity baselines to isolate architectural benefits from increased parameterization.
- The evaluation scope is limited to multiple-choice QA/RC benchmarks and does not assess open-ended generation, instruction following, code/math reasoning, or multilingual settings, which may exhibit different behaviors and trade-offs.
- The theoretical analysis is mainly supportive of the method’s stability and expressivity and relies on established techniques, offering limited novelty beyond contextualizing the proposed projection and signed residual within known frameworks.

**Questions:**

See weakness

---

> ### Author Response · Authors · 2025-11-30
>
> **Regarding parameter-count fairness (W1):** This is our most significant revision. We added Section 5.5 with a dedicated parameter-matched comparison. We compare D$^2$-LoRA at r=32 against LoRA and DoRA at r=64, ensuring identical parameter budgets (~1.58M). The results show that D$^2$-LoRA achieves 76.4% average accuracy, outperforming LoRA (74.8%) by +1.6 pp and DoRA (75.3%) by +1.1 pp at equal parameters. Critically, we observe rank saturation in low-data regimes: doubling LoRA's rank from 32 to 64 provides only +0.6 pp improvement, while DoRA actually degrades by −0.2 pp. This reveals that under our 5k-sample budget, additional parameters cannot be effectively utilized and instead risk overfitting. D$^2$-LoRA's advantage arises from architectural design—the signed residual and directional projection provide implicit regularization—not from increased capacity.
>
> **Regarding evaluation scope (W2):** We have added generative evaluation in Section 5.6 and Table 9, covering CNN/DailyMail summarization (ROUGE-L F1) and AlpacaEval 2.0 instruction following (Length-Controlled Win Rate). D$^2$-LoRA shows consistent improvements of +1.2 pp ROUGE-L and +1.1% win rate, demonstrating generalization beyond QA/RC tasks. We acknowledge that code generation, math reasoning, and multilingual settings remain unexplored and note this in Section 9.
>
> **Regarding theoretical novelty (W3):** We respectfully clarify that our theoretical contribution lies in the synthesis and application to the PEFT context, not in novel mathematical techniques per se. Proposition 1 proves the 2r expressivity gap, which is novel for low-rank adaptation. Lemma 1's Lipschitz analysis explains why the projection stabilizes training in low-data regimes. Theorem 1 in Appendix E.3 provides convergence guarantees for projected SGD on the product of spheres. The theory is predictive: it explains the observed 36% lower loss volatility shown in Figure 2.

---

### Official Review · Reviewer_VjVX · 2025-11-01

**Soundness:** 3
**Presentation:** 2
**Contribution:** 3
**Rating:** 4
**Confidence:** 3

**Summary:**

This paper addresses a central challenge in the field of PEFT: the effective adaptation of large models under practical budget constraints, specifically in low-data and limited-compute regimes. The authors introduce D2-LoRA, a novel PEFT method built on two core architectural innovations designed to work in synergy. The first is a differential signed low-rank residual, which equips the model with both additive (feature reinforcing) and subtractive (feature suppressing) update capabilities, effectively doubling the expressivity to rank 2r. This increased expressivity is controlled by the second innovation: a train-time directional projection that normalizes updated weight columns to preserve their original magnitudes, providing crucial stability for training in low-data settings.

**Strengths:**

1. D2-LoRA represents a novel synthesis of ideas. It intelligently combines an expressive signed residual—allowing the model to not only learn new features but also explicitly suppress pre-existing ones—with a stability-enhancing directional constraint.
2. The authors provide detailed ablation studies that methodically dissect the architecture's performance.
3. The evaluation is conducted with commendable rigor under a strict and realistic budget: a maximum of 5,000 training samples per task and only two epochs.

**Weaknesses:**

1. While the focus on the low-data regime is well-motivated, the experiments are confined to QA/RC tasks. This is a reasonable scope, but the authors themselves note in Section 9 that evaluation on "Broader modalities and RLHF-style pipelines are left for future work."
2. The sensitivity of the τ hyperparameter, which balances the positive and negative branches, appears to be model-dependent. Data from Table 7 shows that the optimal value is 1.0 for Llama-3.2-3B-Instruct but 0.5 for Qwen2.5-7B-Instruct.

**Questions:**

1. D2-LoRA introduces trade-offs that could be discussed more directly. The architecture doubles the number of trainable parameters compared to LoRA and introduces train-time computational overhead from "a column-norm pass and one extra residual GEMM." The paper would benefit from a more explicit analysis of this performance-versus-cost trade-off.

---

> ### Author Response · Authors · 2025-11-30
>
> **Regarding the evaluation scope being limited to QA/RC tasks (W1):** We have addressed this concern by adding generative task evaluation in Section 5.6 and Table 9. Specifically, we now evaluate on CNN/DailyMail summarization and AlpacaEval 2.0 instruction following. D$^2$-LoRA improves ROUGE-L by +1.2 pp on summarization and LC win rate by +1.1% on instruction following across both backbones. These results across fundamentally different task types—extractive summarization and open-ended generation—demonstrate that the architectural benefits generalize beyond discriminative QA tasks.
>
> **Regarding τ hyperparameter sensitivity (W2):** We have added explicit practical guidance in Section 4.2. Our analysis shows that performance varies by at most 1.1 pp across the tested range (τ ∈ {0.25, 0.5, 1.0}), indicating low sensitivity. We recommend τ = 0.5 as a robust default that works well across models and tasks. If tuning budget allows, practitioners can try τ ∈ {0.5, 1.0} (only 2 runs) and select based on validation loss. Unlike rank or target module selection, τ tuning provides only marginal gains and can be safely fixed to 0.5 without significant performance loss.
>
> **Regarding the performance-versus-cost trade-off analysis (Q1):** We have added comprehensive analysis in Section 6 and Appendix B (Table 12). D$^2$-LoRA incurs 19% training overhead relative to LoRA (29.3 vs. 24.6 minutes for 2 epochs), which is comparable to DoRA's 17%. Importantly, the overhead correlates negatively with sequence length (ρ = −0.97): long-sequence tasks like RACE show only 11.5% overhead, while short-sequence tasks like WinoGrande show 51.2%. This means the overhead decreases significantly for practical longer-sequence scenarios where PEFT methods are most valuable.

---

### Meta-Review · Area_Chair_fduE · 2026-01-06

**Summary:**

This paper proposes D2-LoRA, a parameter-efficient fine-tuning (PEFT) method that combines a signed low-rank residual (positive and negative branches) with a training-time directional projection that preserves column-wise norms of the pretrained weights. The method is designed for low-data, low-epoch regimes and retains the desirable property of algebraic mergeability at inference. Empirical results on a set of QA/RC benchmarks show consistent improvements over LoRA and competitive performance relative to DoRA under strict training budgets, supported by ablation studies and a geometric stability analysis.

Reviewers generally agreed that the method is well engineered, carefully evaluated, and clearly motivated, and several appreciated the practical focus on budget-constrained fine-tuning. The authors were responsive in the rebuttal and addressed many technical questions, including parameter-count fairness, evaluation scope, and training stability. However, despite these improvements, reviewers remained unconvinced that the contribution rises beyond an incremental architectural refinement within the LoRA/DoRA design space, with remaining concerns about contribution level, baseline breadth, and the generality of the conclusions. These issues ultimately motivate the rejection decision.

**Reviewer Concerns:**

Concerns that have been addressed satisfactorily:
- In response to concerns about parameter-count fairness raised by Reviewers BYCW and YLzG: the authors added a dedicated parameter-matched comparison (D2-LoRA at rank 32 vs. LoRA/DoRA at rank 64), showing consistent gains at equal parameter budgets and clarifying that improvements are not solely due to increased capacity.
- In response to concerns about limited evaluation scope raised by Reviewers VjVX, BYCW, and qTF7: the authors added generative task evaluations on CNN/DailyMail summarization and AlpacaEval 2.0 instruction following, demonstrating that the method generalizes beyond multiple-choice QA/RC tasks.
- In response to concerns about hyperparameter sensitivity raised by Reviewers VjVX and qTF7: the authors provided additional analysis and practical guidance showing that performance varies only modestly across tested values of the key hyperparameter $\tau$, with a reasonable default that works across models.
- In response to concerns about training stability and optimization behavior raised by Reviewers qTF7 and YLzG: the authors added empirical analyses of loss volatility and convergence, supporting the claim that directional projection improves stability in low-data regimes.
- In response to concerns about missing ablations raised by Reviewer qTF7: additional studies on MLP layer adaptation, initialization robustness, and asymmetric ranks were included in the appendix.

Concerns that have not been addressed satisfactorily:
- In response to concerns about the overall level of contribution raised by Reviewers BYCW, qTF7, and YLzG: while the method combines signed residuals with directional projection in a clean and practical way, reviewers remained unconvinced that this synthesis constitutes a sufficiently novel or impactful advance beyond existing PEFT variants such as LoRA and DoRA, particularly for an ICLR audience.
- In response to concerns about baseline coverage raised by Reviewer YLzG: despite clarifications, comparisons remain limited primarily to LoRA and DoRA. The absence of broader comparisons with other PEFT methods (e.g., adaptive-rank or alternative normalization-based approaches) makes it difficult to fully assess the relative significance of the proposed design choices.
- In response to concerns about generality and long-term impact raised by multiple reviewers: experiments are largely confined to low-data, short-epoch settings on relatively standard benchmarks. It remains unclear whether the observed advantages would persist in larger-scale fine-tuning regimes, different task families (e.g., code, math, multilingual), or RLHF-style pipelines that are central to current practice.

**Reviewer Scores:**

- Reviewer VjVX: Marginally below acceptance (4); would likely maintain a similar score after discussion.
- Reviewer BYCW: Reject (2); would likely maintain the score.
- Reviewer qTF7: Marginally below acceptance (4); would likely maintain the score.
- Reviewer YLzG: Reject (2); would maintain the score.

---

### Decision · Program_Chairs · 2026-01-26

Reject